# Mutations in Exons 8 and 11 of *c-kit* Gene in Canine Subcutaneous Mast Cell Tumors and Their Association with Cell Proliferation

**DOI:** 10.3390/vetsci9090493

**Published:** 2022-09-10

**Authors:** Polly Chen, Laura Marconato, Silvia Sabattini, Matti Kiupel

**Affiliations:** 1College of Veterinary Medicine, Michigan State University, East Lansing, MI 48824, USA; 2Department of Veterinary Medical Sciences, University of Bologna, 40126 Bologna, Italy; 3Veterinary Diagnostic Laboratory, Michigan State University, Lansing, MI 48910, USA

**Keywords:** mast cell tumor, subcutaneous, *c-kit*, CD117, mutation, canine, dog, prognosis receptor tyrosine kinase, AgNOR, Ki67, mitotic count

## Abstract

**Simple Summary:**

Mast cell tumors (MCTs) are one of the most common skin tumors in dogs with variable clinical behavior ranging from benign lesions to those causing widespread metastasis. Prognostic factors have been intensively studied in cutaneous MCTs but are less commonly investigated in subcutaneous MCTs as the majority are benign. Activating mutations in exons 8 and 11 of *c-kit*, a gene that regulates proliferation and differentiation of mast cells, occur commonly in canine cutaneous MCTs and are strong predictors of prognosis. *c-kit* mutations have rarely been reported in subcutaneous MCTs. The goal of this study was to identify the prevalence of *c-kit* mutations in exons 8 and 11 in 216 canine subcutaneous MCTs and to investigate their association with other prognostic factors, including mitotic count, histologic grade, KIT pattern and proliferation markers. We detected *c-kit* mutations in exons 8 and 11 in 23 (10.6%) and 12 (5.56%) subcutaneous MCTs, respectively. *c-kit* mutations in exon 11 were associated with histologic high grade and a high mitotic count, suggesting that these parameters can predict the biological behavior of subcutaneous MCTs in a similar manner as in their cutaneous counterparts.

**Abstract:**

The prognostic significance of internal tandem duplication (ITD) mutations in exons 8 and 11 of *c-kit* has been well-described for canine cutaneous mast cell tumors (MCTs), but c-kit mutations have rarely been reported in subcutaneous MCTs. The objective of this study was to identify the prevalence of ITD mutations in exons 8 and 11 of *c-kit* in canine subcutaneous MCTs and to investigate its association with histologic grade, KIT pattern, and proliferation markers. ITD mutations in exons 8 and 11 of *c-kit*, mitotic count, Ki67 index, AgNOR number, Ki67xAgNOR score, KIT pattern, and histologic grade (two-tier system) were retrospectively recorded for 216 dogs with subcutaneous MCTs. ITD mutations in exons 8 and 11 of *c-kit* were detected in 23 (10.6%) and 12 (5.56%) subcutaneous MCTs, respectively. Exon 11 mutations were significantly associated with Kiupel high grade (*p* < 0.001) and increased mitotic count (*p* < 0.001) compared to subcutaneous MCTs with no mutations in exons 8 or 11 (*p* = 0.002) or subcutaneous MCTs with a mutation in exon 8 (*p* = 0.001). There was no significant association of either *c-kit* mutation with KIT patterns or proliferation activity. This study identified a higher prevalence of ITD mutations in exons 8 and 11 of *c-kit* in subcutaneous MCTs than previously reported. Like their cutaneous counterpart, subcutaneous MCTs with exon 11 mutations were more likely to be histologically high grade and have a higher mitotic count, whereas such associations were not observed in subcutaneous MCTs with exon 8 mutations.

## 1. Introduction

Mast cell tumors (MCTs) are one of the most common skin tumors in dogs with variable biological behavior, ranging from benign localized masses, for which surgical excision is curative, to aggressive tumors with increased recurrence, metastasis, and mortality [1,2]. Based on their location within the skin, MCTs have been divided into cutaneous and subcutaneous tumors. This anatomical division is important, as more than 90% of subcutaneous MCTs are considered benign and can be controlled with complete surgical excision [3,4,5,6]. However, this still leaves almost 10% of dogs diagnosed with subcutaneous MCTs succumbing to MCT-related disease [6]. Based on the published literature, approximately 8–9% of subcutaneous MCTs will recur, and 4–6% will metastasize [5,6]. Lastly, 11% of dogs with subcutaneous MCTs will develop another MCT [6].

Histologic grading has been the primary tool to prognosticate canine MCTs [1,2,7,8,9]. While the Patnaik grading system applied the histologic grading to both cutaneous and subcutaneous MCTs, the more recent two-tier system has been shown to be prognostically significant for cutaneous MCTs only [1,7,8,9]. However, some criteria that are being evaluated as part of the two-tier grading system, such as a mitotic count (>4 in 10 HPFs) and multinucleated giant cells (at least one cell in 10 HPFs), have been shown to be prognostically significant in subcutaneous MCTs, albeit with different established cut-off values [6,10]. Studies evaluating the prognostic significance of the two-tier system for canine subcutaneous MCTs are lacking. In addition to histologic grading, proliferation markers, such as argyrophilic nucleolar organizing regions (AgNORs) and Ki67, have been successfully utilized to prognosticate subcutaneous MCTs [5,10]. A Ki67 labeling index of 21 and a combined AgNORxKi67 score above 54 have been associated with more aggressive behavior in cutaneous MCTs [1,11].

The tyrosine kinase receptor KIT plays an important role in mast cell proliferation, differentiation, migration, and survival [12,13,14,15]. As such, tyrosine kinase inhibitors are a popular targeted treatment modality in MCT treatment [12,16,17,18]. Aberrant expression of the KIT protein as determined by immunohistochemistry (IHC) has been associated with a negative prognosis [10,11,15]. Activating mutations of predominantly internal tandem duplication (ITD) and, less often, small insertions/deletions of the *c-kit* gene have been identified in exons 8, 9, 11, 12, and 17, with the highest frequency in the first three [9,13,14,15,19,20,21,22,23,24,25]. ITD mutations in exon 11 of *c-kit* have been reported to occur in approximately 20–45% of canine cutaneous MCTs and have been associated with aberrant KIT protein localization, higher grade, and higher proliferation activity as evidenced by a higher Ki67 index and AgNORxKi67 score [9,12,13,15,21,25,26]. Dogs with cutaneous MCTs with ITD mutations in exon 11 of *c-kit* have an increased incidence of recurrent disease, decreased survival times, and a high risk for MCT-related mortality [2,15,27]. This is in stark contrast to ITD mutations in exon 8 of *c-kit*, which have been detected in up to 33% of canine cutaneous MCTs but have not been associated with poor prognosis [1,13,27]. Mast cell tumors with exon 8 *c-kit* mutations were associated with longer overall survival times in dogs with cutaneous MCTs than those without exon 8 or 11 mutations [1]. MCTs with exon 8 mutations of *c-kit* had a lower histologic grade and proliferation activity and, less often, an aberrant KIT localization [27].

Interestingly, *c-kit* mutations have rarely been reported in canine subcutaneous MCTs [22,24]. An ITD mutation in exon 8 has recently been reported in a single subcutaneous MCT [22], while previous studies did not detect such mutations [9,10].

## 2. Materials and Methods

### 2.1. Case Selection

The study population was comprised of subcutaneous MCTs submitted as surgical biopsies to the Michigan State University (MSU) Veterinary Diagnostic Laboratory (VDL) between April 2011 and December 2019. Inclusion criteria required the submitted MCT to be located in the subcutis with no involvement of the overlying dermis or deeper fascial planes. All subcutaneous MCTs included in this study were submitted for a prognostic MCT panel that included histologic grading, KIT expression pattern and Ki67 index both determined by immunohistochemistry, and AgNOR count, as well as the calculated combined AgNORxKi67 score.

### 2.2. Signalment

The breed and age of the dog at histologic diagnosis were recorded for each biopsy specimen.

### 2.3. Histologic Grading

The histologic grade was recorded even though it has not been validated for subcutaneous MCTs. Histologic grading was performed on HE-stained sections of all 216 MCTs in accordance with the 2-tier grading system by Kiupel et al. for cutaneous MCTs [7]. A high histologic grade was assigned to subcutaneous MCT that met at least one of the following published criteria: at least 7 mitotic figures in 10 high-power fields (hpfs, equal to 2.37 mm^2^), or karyomegaly (i.e., nuclear diameters of at least 10% of neoplastic cells vary by at least two-fold), or at least 3 bizarre nuclei in 10 hpfs, or at least 3 multinucleated (3 or more nuclei) cells in 10 hpfs [7]. MCTs that did not meet these criteria were diagnosed as histologically low grade.

### 2.4. KIT Expression Patterns

Cellular localization of the KIT protein was determined by IHC labeling as previously described [11]. Cellular localization patterns are classified as follows: 1. KIT pattern I: predominantly membranous labeling; 2. KIT pattern II: focal to stippled cytoplasmic labeling with decreased membrane-associated labeling; and 3. KIT pattern III: diffuse cytoplasmic labeling [1,11].

### 2.5. Ki67 Index, AgNOR Count, and Combined AgNOR x Ki67 Score

IHC for Ki67 was determined by IHC labeling for the Ki67 as previously described [11], and Ki67-positive cells were quantified as the average number of positively labeled neoplastic nuclei per area of a 1 cm^2^ optical grid reticle at a magnification of 40× (5 grid areas counted) in the highest labeling area [1,11]. Histochemical staining for AgNORs was performed as previously described [1,11]. AgNORs were counted in 100 randomly selected neoplastic mast cells throughout the tumor at 1000× magnification. Individual AgNORs were resolved by focusing up and down while counting within individual nuclei. Average AgNOR counts/cells were then determined on the basis of averaging the counts within these 100 random neoplastic cells [1,11]. Quantification of tumor proliferation (combined AgNOR x Ki67 score) was performed by multiplying the Ki67 index with the AgNOR count.

### 2.6. Screening for Mutations in Exons 8 and 11 of c-kit

ITD mutations in exons 8 and 11 of c-kit were determined by polymerase chain reaction (PCR) using previously described primer pairs [19]. For exon 8, primers were located in introns 7 to 8 and amplified the previously reported 12-bp duplication mutation in canine MCTs [13,27]. For exon 11, primers flanked exon 11 and the 5′ end of intron 11, which amplifies the previously described region of the ITD mutation in canine MCTs [14,15]. Amplifications for both primer sets were performed using the Type-it Mutation Detection PCR Kit (Qiagen) as previously described [15,27]. PCR products were visualized on the QIAxcel Capillary Electrophoresis System (Qiagen) [19].

### 2.7. Statistical Analyses

Categorical variables were summarized as frequency (percentage), and numerical variables were summarized as median (range). For each case, the histologic grade (2-tier Kiupel system), mitotic count, Ki67 index, AgNOR numbers, combined AgNOR x Ki67 score, KIT pattern (1, 2, 3), and *c-kit* mutation status for exons 8 and 11 were determined. The distribution of the aforementioned tumor markers and signalment were compared in subcutaneous MCTs with or without c-kit mutations. Specifically, MCTs with exon 8 and exon 11 mutations were compared to each other and to subcutaneous MCTs without exon 8 or 11 mutations.

Mann–Whitney U test was used to determine differences in continuous variables (i.e., age, mitotic count, Ki67 index, AgNOR numbers, and AgNOR x Ki67 score), whereas categorical variables (i.e., histologic grade, breed, and KIT pattern) were assessed by means of chi-squared/Fisher’s exact test. Significance was set at *p* < 0.05.

## 3. Results

Two hundred and sixteen samples diagnosed as subcutaneous MCTs were included in this study. Of these, breeds included 67 mixed-breed dogs, 37 Labrador Retrievers, 12 Golden Retrievers, 7 Boxers, 7 Pugs, 6 Beagles, 6 Shih Tzu dogs, 5 American Pit Bull Terriers, 3 Boston Terriers, 3 Chihuahuas, 3 German Shorthaired Pointers, 3 Maltese dogs, 3 Siberian Huskies, 3 Vizslas, 2 American Bulldogs, 2 Cocker Spaniels, 2 Dachshunds, 2 Doberman Pinschers, 2 English Setters, 2 French Bulldogs, 2 Great Pyrenees, 2 Havanese, 2 Miniature Schnauzers, 2 Rhodesian Ridgebacks, 2 Staffordshire Terriers, 1 Australian Shepherd, 1 Belgian Malinois, 1 Bernese Mountain Dog, 1 Blue Heeler, 1 Bullmastiff, 1 Coton de Tulear, 1 Fox Terrier, 1 Jack Russell Terrier, 1 Toy Poodle, 1 Portuguese Water Dog, 1 Schnauzer, 1 Shetland Sheepdog, 1 Tibetan Terrier, 1 Weimaraner, 1 Yorkshire Terrier, and 14 unspecified others.

ITD mutations of *c-kit* were detected in exon 11 in 12/216 (5.6%) subcutaneous MCTs and in exon 8 in 23/216 (10.6%) cases. No mutations in exons 8 or 11 were detected in 181/216 (83.8%) subcutaneous MCTs. No tumors in this study had a concurrent mutation in exons 8 and 11.

Labrador Retrievers had a high prevalence of subcutaneous MCTs with *c-kit* mutations, with 8 of 37 (22%) MCTs having exon 8 mutations and 4 of 37 (11%) MCTs having exon 11 mutations. A distinction between black Labradors and yellow Labradors was not available from this database. No other breed predilections were identified with our sample size.

The median age of dogs with subcutaneous MCTs with *c-kit* mutations in exon 8 was 5 years, which was significantly lower than the median age of dogs with no mutations in exons 8 or 11 (8 years, *p* = 0.001). Dogs with subcutaneous MCTs with *c-kit* mutations in exon 11 also had a significantly higher median age (9 years, *p* = 0.017).

There were 173 (80.1%) low-grade and 43 (19.9%) high-grade tumors. The proportion of high-grade subcutaneous MCTs was significantly higher in tumors with c-kit mutations in exon 11 (8/12; 66.7%), compared to those with exon 8 mutations (3/23; 13%; *p* = 0.002) or to MCTs with no c-kit mutations in exons 8 and 11 (32/181; 17.6%; *p* < 0.001; Figure 1).

The median mitotic count was 1 for both low-grade and high-grade subcutaneous MCTs tumors (range, 0–49 for high-grade and 0–6 for low-grade). Of the high-grade subcutaneous MCTs, 29/43 (67.4%) had a mitotic count of 7 or higher, while none of the low-grade tumors had a count of 7 or higher. Exon 11 *c-kit* mutations were also associated with an increased mitotic count (median mitotic count, 2, range 1–10; Figure 2) compared to mutations in exon 8 (median mitotic count, 1, range 0–11; *p* = 0.001) or dogs with no mutations in exons 8 or 11 (median mitotic count, 1, range 0–49; *p* < 0.001). There were no statistically significant differences in histologic grades between subcutaneous MCTs with a mutation in exon 8 and no mutations in exons 8 or 11.

There were no statistically significant differences in proliferation markers (Ki67 index, AgNOR count, AgNOR x Ki67 index) or in KIT protein localization patterns in subcutaneous MCTs with mutations in either exon 8 or exon 11 when compared to each other or to subcutaneous MCTs without exon 8 or 11 mutations (Figure 3). Of the high-grade subcutaneous MCTs, 23/43 (53.5%) had a Ki67 index of 23 or higher and a combined AgNOR x Ki67 index of 54 or higher. Only 19/173 (10.9%) low-grade subcutaneous MCTs had a Ki67 index of 23 or higher.

## 4. Discussion

This is the first large-scale study investigating the *c-kit* mutation status in canine subcutaneous MCTs. In contrast to previous studies, we identified ITD *c-kit* mutations in exon 11 and exon 8 in 5.6% and 10.6% of examined cases, respectively. Compared to the study population, Labrador Retrievers had a higher prevalence at 11% and 22%, respectively. As Labrador Retrievers have been reported to have a higher risk for developing MCTs and low-grade MCTs in particular, larger-scale studies are necessary to determine whether MCTs in this breed have a higher prevalence of c-kit mutations than MCTs in other breeds [28,29]. The high prevalence of subcutaneous MCTs with a mutation in exon 8 is especially unusual, as approximately 4% of cutaneous MCTs carry this mutation [13]. Furthermore, the prevalence of subcutaneous MCTs with either mutation was less than half the prevalence that has been reported for cutaneous MCTs [1,13,15,19,27]. Interestingly, the ratio of subcutaneous MCTs with a mutation in exon 11 compared to subcutaneous MCTs with a mutation in exon 8 (ratio 1:2) was inversely related to what has been reported for cutaneous MCTs (ratio 2:4:1) [1,27]. The relatively lower number of subcutaneous MCTs with a mutation in exon 11 correlates with the less aggressive biological behavior of these tumors, as ITD mutations in exon 11 of *c-kit* have been associated with a poor prognosis for dogs with cutaneous MCTs [2,6,9,10,15,27].

While the current study lacked clinical outcome data, both a high mitotic count as well as a high histologic grade based on the two-tier grading system were significantly associated with exon 11 mutations, while for subcutaneous MCTs with no mutations in exons 8 and 11 or with a mutation in exon 8, no such association was detected. As an increased mitotic count has been demonstrated to predict higher local recurrence and metastasis in subcutaneous MCTs [5,6,10], we surmise that the detection of an ITD *c-kit* mutation in exon 11 also indicates a more aggressive biological behavior in subcutaneous MCTs, similar to their cutaneous counterpart.

Both a high mitotic count and an exon 11 mutation were significantly associated with a higher grade. A prospective study should confirm the prognostic significance of the two-tier grading system and the detection of ITD mutations in exon 11 of *c-kit* in subcutaneous MCTs. A total of 8/43 (18.60%) subcutaneous MCTs classified as high-grade had an exon 11 ITD mutation. The prevalence of exon 11 mutations tends to be much higher in high-grade cutaneous MCTs [9,21,26], and a recent study reported such a mutation in 62/75 (82%) cutaneous MCTs [27]. The much lower prevalence of aggressive subcutaneous MCTs compared to high-grade cutaneous MCTs and the lower prevalence of such high-grade subcutaneous MCTs to carry an exon 11 mutation may be responsible for previous studies not being able to identify exon 11 mutations in subcutaneous MCTs [10]. While 32/43 (74.4%) subcutaneous MCTs classified as high-grade had no mutations in exons 8 and 11, a mitotic count above 7 in 67.4% of tumors in this group still supports an aggressive behavior [6,10]. These data may also suggest that exon 11 mutations of *c-kit* are less often the driver of aggressive behavior in canine subcutaneous MCTs compared to cutaneous MCTs. Interestingly, only 53.5% of these high-grade subcutaneous MCTs had a Ki67 index above 23 or an AgNORxKi67 index above 54. The larger number of cases with a high mitotic count than those with a Ki67 index above the threshold established for cutaneous MCTs may reflect karyologic abnormalities rather than an increased proliferation activity. Regardless, a prospective study will be necessary to fully establish the prognostic significance of histologic grading for subcutaneous MCTs.

Exon 11 *c-kit* mutations play a crucial role in the oncogenesis of cutaneous MCTs, especially in mast cell proliferation, and we expected that the various proliferation markers investigated in this study would be significantly higher in subcutaneous MCTs with a mutation in exon 11 compared to subcutaneous MCTs with no mutation in exons 8 and 11, similar to what has been reported for cutaneous MCTs [1,11,27]. Furthermore, the aberrant expression of KIT is more commonly observed in cutaneous MCTs with an exon 11 mutation [6,15,19,27,30]. Neither finding could be confirmed for the subcutaneous MCTs with an exon 11 mutation in this study. As we detected only 12 subcutaneous MCTs with such a mutation, the low number may have negatively impacted statistical analysis.

Similar to recently published data for cutaneous MCTs, a mutation in exon 8 of *c-kit* was not associated with any parameter investigated in this study that predicts a poor prognosis [27]. Based on these previously published data and the data presented here, it seems unlikely that a *c-kit* mutation in exon 8 causes a gain-of-function of KIT that would thereby stimulate increased tumor proliferation.

Although the overall number of mutations detected in exons 8 and 11 of *c-kit* in subcutaneous MCTs was low, this retrospective study suggests a prognostic significance for detecting mutations. Furthermore, the histologic grade using the two-tier system may also be helpful in identifying subcutaneous mast cell tumors with a more aggressive biological behavior. Future prospective studies investigating the role of *c-kit* mutations and histologic grade in subcutaneous MCTs for predicting clinical disease progression and risk for metastatic disease and MCT-associated mortality are needed to confirm these hypotheses.

## 5. Conclusions

Mutations in exon 8 and 11 of *c-kit* have been reported in up to 45% of canine cutaneous MCTs [9,13,14,15,19,20,21,22,24,25,27], but only a single case of an exon 8 mutation has been reported in a subcutaneous MCT [22]. This study of 216 dogs identified a higher than anticipated prevalence of approximately 11% and 6% of ITD mutations in exons 8 and 11 of *c-kit* in subcutaneous MCTs, respectively. Similar to cutaneous MCTs, our study demonstrated that *c-kit* mutations exon 11 in subcutaneous MCTs were significantly associated with a histologic high grade (*p* < 0.001) and an increased mitotic count (*p* < 0.001) compared to subcutaneous MCTs with no *c-kit* mutations in exons 8 or 11 (*p* = 0.002) or subcutaneous MCTs with a *c-kit* mutation in exon 8 (*p* = 0.001). The larger number of exon 8 then exon 11 *c-kit* mutations in subcutaneous MCTs represents an inverse relationship compared to cutaneous MCTs that correlates with the less aggressive biological behavior of subcutaneous MCTs. There was no significant association of either *c-kit* mutation with KIT patterns or proliferation activity. These results provide future directions for prospective studies to confirm *c-kit* mutations in exon 11 and a histologic high grade as prognostic factors for subcutaneous MCTs.

## Figures and Tables

**Figure 1 vetsci-09-00493-f001:**
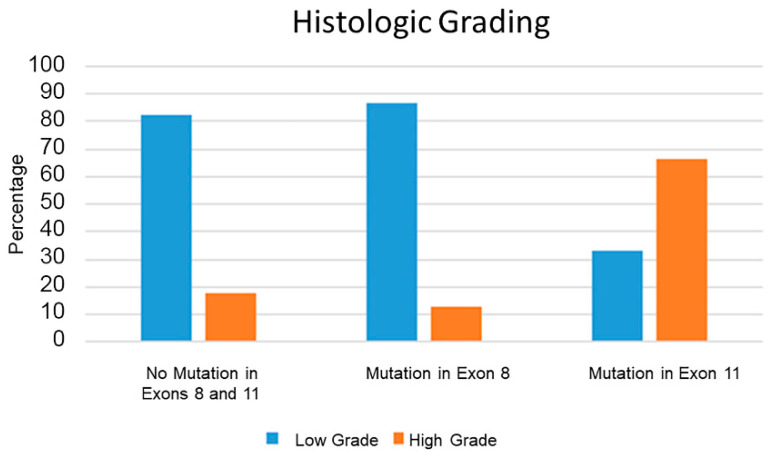
Histogram of percentage of low- (blue) and high-grade (orange) subcutaneous mast cell tumors associated with each of the three groups of different mutation status. A high grade was significantly associated with an internal tandem duplication mutation in exon 11 (*p* < 0.001), but not with a mutation in exon 8 or with no mutations in exons 8 and 11.

**Figure 2 vetsci-09-00493-f002:**
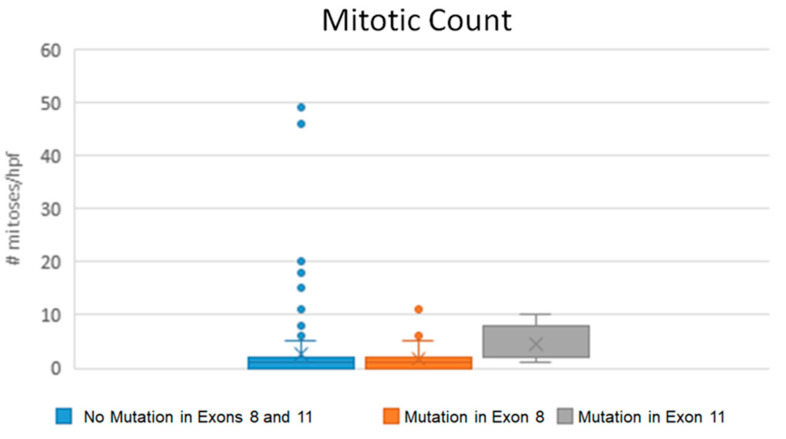
Boxplot of mitotic count per 10 high-power fields in subcutaneous mast cell tumors with different mutation status. Internal tandem duplication mutations in exon 11 (gray) were significantly associated with a higher mitotic count compared to exon 8 (orange) mutations (*p* = 0.001) and no (blue) mutations (*p* < 0.001), but there was no difference between the latter two groups.

**Figure 3 vetsci-09-00493-f003:**
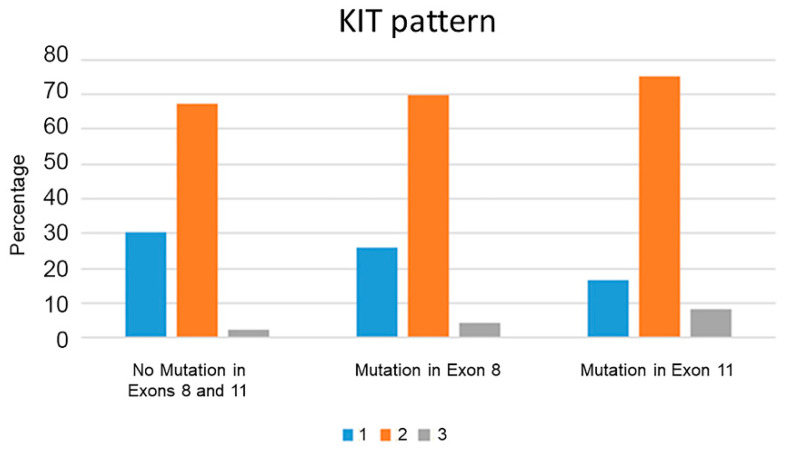
Histogram of percentage of KIT patterns (1 (perimembranous; blue), 2 (focal or stippled cytoplasmic; orange), and 3 (diffuse cytoplasmic, gray)) in subcutaneous mast cell tumors with different mutation status. While there was an overall increase in aberrant labeling (patterns 2 and 3) in mast cell tumors with an internal tandem duplication mutation in exon 11, there was no statistically significant difference.

## Data Availability

The data presented in this study are available on request from the corresponding author.

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
