# Peer review of "Mutations in Exons 8 and 11 of c-kit Gene in Canine Subcutaneous Mast Cell Tumors and Their Association with Cell Proliferation"

_vetsci, 2022, doi:10.3390/vetsci9090493_

Round 1

Reviewer 1 Report

1) Brief Summary

This work contributes to the prognosis of subcutaneous mast cell tumors (MCT) in dogs. The retrospective study aimed to identify the prevalence of tandem mutations in exons 8 and 11 of c-kit in canine subcutaneous MCT and their association with histologic grade, KIT pattern, and proliferation markers. The main finding was that exon 11 mutation was significantly associated with high histological grading and increased mitotic count. However, neither mutation affected cell proliferation, KIT patterns, or proliferation markers AgNORs and Ki67. However, a few changes and a fresh approach to the data could further improve the merit of this paper.

2) General concept comments

I. The Introduction is clear and appropriate.

II. The Materials and Methods are concise and adequate.

III. Results

            I have a few suggestions and questions for the authors.

a) In the second paragraph of the "Results" section, the authors mention the numbers of high and low-grade tumors. Then, three paragraphs with another kind of data follows, and at the end of the page, just before Figure 1, high and low-grade tumor data is mentioned again. It would be more comprehensive if the authors rearranged the order of the paragraphs.

b) The figures are not self-explanatory. It is difficult to figure out the data displayed and its statistical significance without peeking at the text. The mitotic count data is very unclear and is pivotal information used in the discussion. Please provide more details on the figure.

c) I would be more careful to declare that "Labrador retrievers had the highest prevalence of subcutaneous MCTs with c-kit mutations, with 8 of 37 (22%) MCTs having exon eight mutations and 4 of 37 (11%) MCTs having exon 11 mutations". When the author put this way, it seems like a fair comparison (which was not) with the other breeds they obtained samples from, even by adding a "disclaimer" at the end of the paragraph (No other breed predilection was identified within our sample size). However, if the authors changed the sentence to "Labrador retrievers had a high prevalence of subcutaneous MCTs...", it would seem more appropriate as far as methodology is concerned and a piece of valuable prognostic information.

d) When you crunch the numbers, there is another angle to the results. From the total of 216 samples, 43 were classified as high-grade MCTs. A total of 8/43 (18.60%) showed exon 11 ITD mutations; 3/43 (6.98%) had exon eight mutations, and a whopping 149 (74.42%) showed mutations of neither exons. Do the authors believe the relationship between the 2-tier grading system results and the frequency of the c-kit ITD mutations is relevant? If so, were those numbers expected? Is it worth adding them to the results and discussion in your paper?

IV. Discussion

            I also have a few suggestions and questions for the authors.

a) Please consider rephrasing the sentence "Labrador Retrievers had a much higher prevalence at 11% and 22%, respectively", for the reasons mentioned above.

b) The authors claim in the discussion: "While the current study lacked clinical outcome data, both a high mitotic count as well as a high histologic grade, based on the two-tier grading system, were significantly associated with exon 11 mutations, while for subcutaneous MCTs with no mutations in exons 8 and 11 or with a mutation in exon eight no such association was detected." As discussed above, this looks irrefutable considering the number of high-grade samples among MCTs with c-kit exon 11 ITD mutations (8/12). On the other hand, from the total of 43 high-grade MCTs, the proportion of samples with no mutations in exons 8 and 11 (32/43) exceeds by a 4:1 ratio the number of cases with exon 11 ITD mutations (8/43). Does it not configure an association between the two-tier grading system and a high absence index of exons 8 and 11 mutations?

c) "Although the overall number of c-kit mutations detected in exons 8 and 11 in subcutaneous MCTs was low, this retrospective study suggests a prognostic significance for detecting mutations, especially in exon 11." The number of exon 11 mutations was meager; four (1/3) were verified in Labrador Retrievers. The sentence should end with the word mutations, in my opinion.

d) "Furthermore, the histologic grade, using the two-tier system, may also help identify subcutaneous mast cell tumors with a more aggressive biological behavior". I agree, particularly if the mitotic data in the text becomes clearer.

Author Response

Thank you for you thorough review and very helpful suggestions, we appreciate it. Please find the responses to your specific comments below:

a) In the second paragraph of the "Results" section, the authors mention the numbers of high and low-grade tumors. Then, three paragraphs with another kind of data follows, and at the end of the page, just before Figure 1, high and low-grade tumor data is mentioned again. It would be more comprehensive if the authors rearranged the order of the paragraphs.

Thank you for the suggestion. We agree with your statement and rearranged the paragraphs and believe that this helps the flow of the manuscript.

b) The figures are not self-explanatory. It is difficult to figure out the data displayed and its statistical significance without peeking at the text. The mitotic count data is very unclear and is pivotal information used in the discussion. Please provide more details on the figure.

We added more detail to the figure legends and added a figure for the mitotic count.

c) I would be more careful to declare that "Labrador retrievers had the highest prevalence of subcutaneous MCTs with c-kit mutations, with 8 of 37 (22%) MCTs having exon eight mutations and 4 of 37 (11%) MCTs having exon 11 mutations". When the author put this way, it seems like a fair comparison (which was not) with the other breeds they obtained samples from, even by adding a "disclaimer" at the end of the paragraph (No other breed predilection was identified within our sample size). However, if the authors changed the sentence to "Labrador retrievers had a high prevalence of subcutaneous MCTs...", it would seem more appropriate as far as methodology is concerned and a piece of valuable prognostic information.

We changed the wording as suggested.

d) When you crunch the numbers, there is another angle to the results. From the total of 216 samples, 43 were classified as high-grade MCTs. A total of 8/43 (18.60%) showed exon 11 ITD mutations; 3/43 (6.98%) had exon eight mutations, and a whopping 149 (74.42%) showed mutations of neither exons. Do the authors believe the relationship between the 2-tier grading system results and the frequency of the c-kit ITD mutations is relevant? If so, were those numbers expected? Is it worth adding them to the results and discussion in your paper?

This is a great point, thank you for this observation. We had analyzed these data, but for whatever reason decided not to include them in the paper to focus on the difference in mutation. When rearranging the paragraphs, it made perfect sense to include additional information and especially discussion about the high versus low grade MCTs. This data has been added. The prevalence of exon 11 mutations is higher in subcutaneous MCTs classified as high grade than those classified as low grade MCTs, but it is much lower then what we observe in cutaneous high grade MCTs. As a significant number of such high grade subcutaneous MCTs without exon 11 mutation still have a high proliferation activity (both mitotic count and Ki67/AgNOR) one can speculate that c-kit is less commonly a driver of aggressive behavior in subcutaneous MCTs then in cutaneous MCTs. Ultimately we need a prospective study to show efficacy of the grading system for subcutaneous MCTs.

IV. Discussion

I also have a few suggestions and questions for the authors.

a) Please consider rephrasing the sentence "Labrador Retrievers had a much higher prevalence at 11% and 22%, respectively", for the reasons mentioned above.

The sentence was changed as requested.

b) The authors claim in the discussion: "While the current study lacked clinical outcome data, both a high mitotic count as well as a high histologic grade, based on the two-tier grading system, were significantly associated with exon 11 mutations, while for subcutaneous MCTs with no mutations in exons 8 and 11 or with a mutation in exon eight no such association was detected." As discussed above, this looks irrefutable considering the number of high-grade samples among MCTs with c-kit exon 11 ITD mutations (8/12). On the other hand, from the total of 43 high-grade MCTs, the proportion of samples with no mutations in exons 8 and 11 (32/43) exceeds by a 4:1 ratio the number of cases with exon 11 ITD mutations (8/43). Does it not configure an association between the two-tier grading system and a high absence index of exons 8 and 11 mutations?

As mentioned earlier, this is a great comment and we added additional data in results and a paragraph in the discussion to address it.

c) "Although the overall number of c-kit mutations detected in exons 8 and 11 in subcutaneous MCTs was low, this retrospective study suggests a prognostic significance for detecting mutations, especially in exon 11." The number of exon 11 mutations was meager; four (1/3) were verified in Labrador Retrievers. The sentence should end with the word mutations, in my opinion.

We followed the reviewers suggestion.

d) "Furthermore, the histologic grade, using the two-tier system, may also help identify subcutaneous mast cell tumors with a more aggressive biological behavior". I agree, particularly if the mitotic data in the text becomes clearer.

We hope we clarified the mitotic data, thanks again for the helpful comments.

Reviewer 2 Report

This article exposes the importance of mutations in exon 8 and 11 of the c-Kit gene in subcutaneous mast cell tumors. It is a well-written robust and organized article, with interesting and valuable information, and its publication is important for clinicians and pathologists knowledge.

.

Author Response

Thank you for your comments.

Author Response

Thank you for your review and helpful comments. Please find below our responses to the individual comments and suggestions.

The line number is missing, making difficult to comment on specific part of the manuscript.

We truly understand this inconvenience, but we submitted the manuscript within the provided template and no line numbers should be included.

“ITD mutations in exons 8 and 11 of c-kit were detected in 23 (10.6%) and 12 (5.56%) subcutaneous MCTs, respectively. Exon 11 mutations were significantly associated with high grade (P < 0.001)”. Please specify.. Kiupel high grade.

We modified the text as requested.

“There were 173 (80.1%) low grade and 43 (19.9%) high grade tumors. Median mitotic count was 1 (range, 0-49)”  . Mitotic count should be reported separately for the low and high grade with the range.

We agree. This is a good suggestion and we expanded the result section, added a graphic for mitotic count and expanded the discussion. We also added additional information for the other proliferation markers for the two different grades.

 “Labrador retrievers had the highest prevalence of subcutaneous MCTs with c-kit mutations, with 8 of 37 (22%) MCTs having exon 8 mutations and 4 of 37 (11%) MCTs having exon 11 mutations”. Anecdotally black Labrador seems to have worst prognosis with SC MCT , can you report if any difference with yellow and black Labrador in exon 11 mutation? If you cannot specify that this was not possible.

Unfortunately the coat color was no identified within our records.

One of the most important findings in this article is the association of c-KIT with an already reported prognostic factor, the MI. More explanation should be given in to the correlation of of c-KiT mutations and MI and a table should be added that visually make easy to interpret this correlation.

Again, we strongly agree and added more details in the result section including a new figure and discussed this association in more detail.

“There were no statistically significant differences in proliferation markers (Ki67 index, AgNOR count, AgNORxKi67 index), or KIT protein localization patterns in subcutaneous MCTs with mutations in either exon 8 or exon 11 when compared to each other or to subcutaneous MCTs without exon 8 or 11 mutations”. From reading this paragraph, it is not very clear if other correlations have been done. Even if not the main aim of the paper, as often MI and Ki-67 are correlated, correlation of MI and Ki-67 should also be investigated, reported and the results discussed. The same should be done for grade and MI as the MI cut off for SC MST is different from cutaneous MCT.

Please expand this paragraph and make it clearer. Assuming you did not find any correlation between all the proliferation markers, this need to be more deeply discussed and possible explanations reported in the discussion.

Similar to the previous statement, we added additional data and further discussed these. Thank you for these comments.